# Milk Composition of Creole Goats Raised at Different Altitudes in an Extensive Production System in Northeast Mexico

**DOI:** 10.3390/ani13111738

**Published:** 2023-05-24

**Authors:** Luz Y. Peña-Avelino, Ivonne Ceballos-Olvera, Gerardo N. Rosales-Martinez, Javier Hernández-Melendez, Jorge Alva-Pérez

**Affiliations:** 1Faculty of Veterinary Medicine and Animal Science “Dr. Norberto Treviño Zapata”, Autonomous University of Tamaulipas, Ciudad Victoria 87000, Mexico; lypena@docentes.uat.edu.mx (L.Y.P.-A.); iceballos@docentes.uat.edu.mx (I.C.-O.); grosales@uat.edu.mx (G.N.R.-M.); 2Faculty of Engineering and Sciences, Autonomous University of Tamaulipas, Centro Universitario Victoria, Ciudad Victoria 87000, Mexico; javhernan@docentes.uat.edu.mx

**Keywords:** goat milk, semiarid rangeland, lactation stage, altitude, fatty acids

## Abstract

**Simple Summary:**

Creole goat milk is a valuable product for human consumption. Its physicochemical characteristics could be influenced by altitude and lactation stage. In this work, we evaluated several chemical and physical conditions of Creole goat milk in semiarid rangeland. The main results indicated that protein and density are influenced by altitude, while fat, freezing point, and pH are influenced by the lactation stage. One of the most nutritional values of goat milk is the type and concentration of fatty acids. In this work, we found that the effect of the agroecological region was minimal on their concentration. The lactation stage affected the composition of medium-chain fatty acids and linoelaidic acid. We concluded that nutrition by altitude and goat genetic background could contribute to milk composition.

**Abstract:**

Goat milk composition is affected by feeding, and in semiarid rangeland, information on Creole goat milk physicochemical composition is lacking. For the fulfillment of this objective, three agroecological regions (AR) considering altitude (lowland 87, highland 779, and mountain 1309 m above sea level) with different botanical compositions were chosen. Every AR analyzed accounted for 30 goat herds, with a total of 90 herds. The results demonstrated that altitude had an influence mainly on density and protein. Milk density increases as altitude increases; conversely, milk protein increases as altitude decreases. On the other hand, in the mountain and lowland ARs, the salts and solids not fat (SNF) percentages were higher compared to that of the highland AR (*p* < 0.05). The freezing point (FP) was higher at highland altitudes compared to that of mountain and lowland ARs (*p* < 0.01). In the milk fatty acids (FA) profile, only the C14:1 value was affected by altitude, whereas goat milk at lowland and mountain altitudes had higher values compared to that at highland altitudes (*p* < 0.05). Additionally, late lactation stage fat, FP, and pH values were higher compared to early lactation values. The opposite effect was observed for salts and SNF. In the FA profile, late lactation values were higher for C10:0 and C8:0 compared to early lactation values. The opposite trend was observed for C18:2n6t. The thrombogenic index was significantly higher at lowland altitudes compared to highland altitudes, and similar to the mountain AR. These goat milk characteristics could be explained as a consequence of animal nutrition, as well as the goat’s meat-type phenotype.

## 1. Introduction

In the last five years, the world goat population increased by 12.4%, with Asia being the principal continent where goat milk is produced. In 2020, the goat population ranked second in total live animal production in the world [1]. Goats are excellent profitable livestock in smallholder farming systems. These animals have harsh climate tolerance, functional recovery capacity from drought, and a good capacity to thrive under low-quality diets [2,3]. Goat milk demand has been rising throughout the world, both in developed and undeveloped countries. Some of the motivations for goat milk consumption include high nutritional content compared to cow’s milk, filling the need for gourmet products (e.g., goat cheese), and easy access to a valuable animal product for poor people [4,5]. Furthermore, goat milk is used as a therapy for gastrointestinal disturbances and has greater digestibility and lower allergenicity than cow milk [4,6].

Goat breed characterization is crucial to understand productivity [3]. In America, the evolution of European goat breeds gives rise to a unique goat genetic resource. This new kind of goat breed (generally named Creole goat) is adapted to local ecological niches [7] and has productivity implications [8]. To understand Creole goat production, the analysis of the ecological niche, climate, and management could determine factors influencing productivity [9,10]. In Mexico, there is scarce information about Creole goat resources and their productivity. It is known that breed, age, vegetal material fed, and body condition are factors that influence milk yield and chemical composition [6,9,11]. The influence of browsing on semiarid rangeland including forage species, cacti, and scrubs on milk quality parameters needs to be investigated to understand the potential benefits for small systems. There is still a lack of information on how fatty acids (FA) could be modified by altitude and different stages of lactation. It is known that the FA composition and protein content of cow’s milk is affected by the agroecological region and altitude [12]. In goats, it was demonstrated that total protein and dry matter in milk were higher in mountainous areas (500–1000 m above sea level [masl]) compared to that of upland areas (300–500 masl) [13]. These studies demonstrate that ecological niches influence milk parameters. The objective of this study was to determine the effects of altitude and lactation stage on Creole goat milk physicochemical parameters in northeast Mexico where the ecological niche is generally semiarid. The information obtained could set out the nutritional values of the goat milk for the benefit of the producers, as well as set the principles for their exploitation.

## 2. Materials and Methods

All procedures have been approved by the Committee and Ethical Consent of the Faculty of Veterinary Medicine and Animal Sciences of the University of Tamaulipas (reference number CBBA_17_009). Three agroecological sites in the northeast of Mexico (Tamaulipas) were selected for this study: lowland (Lo, mean altitude: 87 masl), highland (Hi, mean altitude: 779 masl), and mountain (Mo, mean altitude: 1309). Nine farms were selected from the three different agroecological sites (Table 1). A non-probabilistic sampling method with participating producers was used. Goats are raised for kid goat production (meat) primarily, and pastoralist communities have different farming activities besides goat handling. For milk physicochemical composition, ten goats from each farm were randomly selected. Selection criteria included good health (goats did not have signs of diarrhea, cough, runny nose, poor body condition, opaque coat, lame, or mastitis), 2.5 years average age, and in possession of an identification number. Goat rearing conditions included a minimum of eight hours of free grazing and regular access to clean water.

The temperature and precipitation ranges are 20–24 °C and 500–800 mm, 12–24 °C and 400–1100 mm, and 10–24 °C and 300–1300 mm, for lowland, highland, and mountain agroecological regions (AR), respectively [14,15,16]. The altitude of the municipality head of the lowland was 81 masl, for the highland, 742 masl, and for the mountain, 1162 masl [17]. The Tamaulipas territory is composed of the Chihuahuan arid region and the Tamaulipan semiarid region [18].

The vegetation of the lowland was classified as forest, which contained bushes and grass species such as *Aristida adscensionis*, *Cynodon dactylon*, and *Dichanthium annulatum*. The predominant tree plants in the area were *Celtis pallida*, *Ebenopsis ebano*, and *Prosopis spp. Parkinsonia aculeata* and cacti-like *Acanthocereus tetragonus*, *Cylindropuntia leptocaulis*, and *Opuntia engelmannii* [18]. The vegetation of the highland was classified as *Yucca spp*., *Hechtia glomerata*, *Agave spp*., *Dasylirion spp*., *Flourensia cernua*, *Larrea tridentata*, prickly scrub forest (*Prosopis spp*.), and grassland [19]. The vegetation of the mountain constituted mainly of 63% shrubland, 23% forest, and 2% grassland. The plants that could be found there are similar to the highland and include Pinus, Quercus, *Juniperus flaccida,* and submountain scrub [20].

Milk samples (n = 180) were collected by hand milking once a day (between 08:00–09:00 in the morning) and were collected two times during lactation; at the early stage of lactation on the 10th day after kidding, and at late lactation on the 70th day. The sampling period (the 10th day in the beginning and the 70th day at the end of the milking period) was considered after attending these periods when the producers use the milk for their family nourishment. Two aliquots were obtained of approximately 40 mL from each goat in a 50 mL sterile plastic bottle and kept in a cooler box before being taken to the laboratory for analysis. One aliquot was stored at 4 °C (for chemical and physical composition analysis) and the other at −20 °C for free fatty acids (FA) profile analysis. The chemical and physical composition analysis included density, freezing point (FP), solids not fat (SNF), milk fat, salts, protein, and lactose. The analyses were performed using a milk analyzer (Master Eco, Milkotester Ltd. Belovo, Bulgaria). For pH measurement, a portable pH meter (pHep^®^, Hanna Instruments, Woonsocket, RI, USA) was used. Before measurement, the samples were preheated at 42 °C and then cooled to 30 °C. Subsequently, the milk sample was thoroughly mixed to evenly distribute fat globules and dissolve any milk residue before reading according to the manufacturer’s instructions. All measurements were obtained in triplicate. The commercial goat milk brand (Meyenberg USA brand) was used as a standard reference.

For the FA analysis of the milk samples, fat separation was performed using the protocol described by Feng et al. [21] and modified by Luna et al. [22]. Briefly, the milk samples were thawed at 4 °C and incubated (liquid milk) at 20 °C for 20 min. Afterward, every sample was centrifuged at 17,800× *g* for 30 min. The fat layer was removed, and a consecutive centrifugation step was performed at 19,300× *g* for another 30 min. The fat was placed in a new tube and frozen at −20 °C until analysis. The fatty acid methyl esters (FAME) were obtained according to Palmquist and Jenkins [23], with the modifications of Jenkins [24]. The FA composition was determined by gas chromatography with a flame ionization detector (GC-FID) in an Agilent 6890N gas chromatograph (Agilent Technologies, Santa Clara, CA, USA) with a capillary column (SP-2560 100 m × 0.25 mm, 0.2 mm width, 2560 Supelco, Inc., Bellefonte, PA, USA). Helium was used as a carrier gas. The identification of the FA was made by comparing the retention times of each peak obtained in the chromatogram with a standard of 37 FAME components (Supelco 37 FAME Components). The hypocholesterolemic and hypercholesterolemic fatty acids ratio (HH) was calculated, as described by Osmari et al. [25]. The atherogenic (AI) and thrombogenic (TI) indexes were obtained according to Ulbright and Southgate [26].

The chemical composition and FA profile were analyzed using a completely randomized design with repeated measurements using the SAS MIXED procedure [27] according to the following model:(1)Yijk=μ+Ri+tj+φij+Pk+tPjk+εijkl
where *Yijk* is the response variable, *μ* is the overall mean, *Ri* is the simple replicate effect (*i* = 1, …90), *τj* is the agroecological region effect (*j* = 1, …3), *φij* is the agroecological region replicate effect (Error a), *Pk* is the lactation stage effect (*k* = early, and late), τ*Pjk* is the interaction of agroecological region and lactation stage, and *εijkl* is the random error (Error b).

The fixed effects included three agroecological regions (lowland, highland, and mountain) and two lactation stages (early stage, 10 d; and late stage, 70 d). The random effects were the goat and residual. Differences between treatment means were considered significant when resultant *p*-values were <0.05. The appropriate covariance structure for the analysis was determined by testing different structures, and the one with the negative or near-zero values according to the Akaike and Schwarz criteria was chosen [28].

## 3. Results

The agroecological region (AR), lactation stage (LS), and their interactions on milk composition are shown in Table 2. All the physico-chemical data are depicted in the Appendix A. The lactose content, fat percentage, and the pH of the goat milk did not differ among the ARs (*p* > 0.05). In contrast, milk crude protein concentration was higher in the lowland AR, while density was higher in the mountain AR. Salts and SNF percentages were higher in mountain and lowland ARs compared to the highland AR (*p* < 0.05). In contrast, freezing point (FP) had lower values in mountain and lowland ARs compared to the highland AR (−0.65 °C vs. −0.62 °C). Additionally, fat percentage, FP, and pH were significantly lower in the early LS compared to the late LS. On the other hand, salts and SNF values were significantly higher in early LS compared to late LS. An interaction effect between AR and LS was observed for milk fat and pH (*p* < 0.05).

FA concentrations and lipidic indexes are shown in Table 3. The complete FAMES data are depicted in the Appendix A. In order, the most abundant milk FA were C16:0, C18:1n9c, C10:0, and C14:0. No differences between AR FA contents were observed, except for C14:1. The C14:1 mountain FA value was higher compared to that in the highland AR, and similar to that of the lowland AR FA value. Further, differences in LS FA contents were observed for C8:0, C10:0, and C18:2n6t. Caprylic (C8:0), and capric (C10:0) FA were lower at the early LS compared to the late LS. Meanwhile, linoelaidic FA (C18:2n6t) content was higher at the early LS compared to the late LS. The analysis of short-chain FA (SCFA) contents demonstrated that higher concentrations were present at the late LS compared to the early LS (*p* = 0.016). No interaction effect was observed in the saturated or unsaturated milk FA. In addition, the lipidic indexes showed that neither AR nor LS affected the hypo:hypercholesterolemic acids ratio (HH) or atherogenic index (AI). The Thrombogenic index (TI) had lower values in the highland AR compared to the lowland AR and similar to the mountain AR (*p* < 0.05).

Taken together, these results demonstrated that the chemical composition of Creole goat milk is heterogeneous concerning AR, mainly for protein, salts, SNF, and C14:1 FA concentration. Concerning the lactation period, the results also demonstrated that LS could contribute to differences in the chemical composition of fat, salts, pH, SNF, as well as FA composition (mainly C8:0, C10:0, and C18:2n6t).

## 4. Discussion

Goat milk attributes are influenced by breed, age, health, agroclimatic conditions, seasonality, and feeding management [29,30,31]. The ruminant diet effect on milk FA content has been investigated, showing the advantages of feeding on forages. Some human health advantages are the lower atherogenic and thrombogenic content, among other health benefits [32]. On the other hand, the impact of vegetation resources from dry ecosystems on milk goats is not yet clear. There has been considerable interest in the milk composition and milk FA of dairy goats from local scrubby rangeland, particularly in native pastures due to their high concentration of polyunsaturated fatty acids (PUFA) [11].

The fresh goat milk density range is about 1029–1039 kg/m^3^ [33]. In our study, the density of milk was highest in the mountain AR (1036.8 kg/m^3^) compared to that from the highland AR (1032.5 kg/m^3^) or the lowland AR (1033.8 kg/m^3^). Our mountain milk density results were similar to the native Greek goat breed localized to 1400 masl [34]. Casein and fat are the principal components that give physical characteristics to goat milk, and these components are limited by animal nutrition and diet [35]. For pastoralist communities (comparable to our investigation), goat nutrition is based mainly on grazing. Therefore, our density results could reflect diverse vegetation resources localized at different altitudes. In contrast, goat milk density evaluated by LS did not demonstrate differences. In agreement with our results, Idamokoro et al. [36] did not observe differences in milk density in Nguni (1034.45 kg/m^3^), Boer (1034.71 kg/m^3^), or nondescript goats (1033.09 kg/m^3^) during LS. Instead, Strzalkowska et al. [37] reported that in white Polish goats, the density of early lactation milk was different from late lactation (1025.9 kg/m^3^ and 1029.8 kg/m^3^, respectively). Goat milk density is directly linked to goat milk components, mainly casein and fat, so it is possible that these components did not vary significantly through the lactation stages measured in our work.

Goat milk protein is composed of caseins and whey proteins, and these factors are affected by breed, lactation period, nutrition status, and environmental stages such as altitude [33,38]. Žan et al. [38] observed that in Saanen and Alpine goats, the milk protein concentration decreased as altitude increased. In agreement with the latter work, our results showed that milk protein was higher for the lowland AR (3.84%), with significant differences compared to highland and mountain AR milk protein concentrations (3.43% and 3.58%, respectively *p* = 0.014). On the other hand, Barlowska et al. [13] showed that for Saanen and goats of unknown genetic origin, milk protein content was higher in mountain areas (up to 500 masl) than in upland areas (300 to 500 masl). These differences in our results could be due to feed management (concentrated feed supplementation vs. no supplementation), as well as a production system (dairy goats vs. kid meat production). In our study, milk protein concentration was not altered by LS. Contrary to our results, Kuchtik et al. [39] and Strzalkowska et al. [37] observed that milk protein increased with the progress of lactation. However, it is important to note that in both research works, they analyzed dairy goats with formulated diets. Of interest, the protein content of brown short-haired [39] or Polish white goats [37] was lower at around 60–70 days of lactation compared to that of Creole goats analyzed in this work (2.9% vs. 3.69%). Overall, the results for milk protein observed in this work could be related to the protein content from forage, as well as genetic background. The forage nitrogen is more abundant in the lowland [37], so it is probable that goat milk in the lowland has a higher protein content than at other altitudes. Besides altitude, the goat breed influences the protein content of milk [37].

Lactose is the main carbohydrate of ruminant milk, and in goats, its range is 4.1–4.3% [5,33]. Some differences are observed between breeds and seasons. Mayer and Fiechter [40] reported a range of 4.04–4.46% for lactose in six different breeds in Austria. They also found lactose seasonal variation. We did not observe differences in lactose levels relating to AR or LS. Nevertheless, we found a lactose concentration above 5.0% (similar to sheep’s milk, according to Park [33]). In agreement with our results, Arief et al. [41] found milk lactose concentrations of 5.1 to 5.3 % in Etawa crossbreed dairy goats. As can be seen in the latter, differences observed in lactose milk concentration could be due to breed genetics. Creole goats analyzed in this study had more of a meat than dairy phenotype, and, remarkably, the genetic background could be important for milk lactose concentration.

The salts and SNF values were lower in the highland AR (0.73% and 9.47%, respectively) compared to that in lowland (0.76% and 9.69%) and mountain ARs (0.76% and 9.89%). Salts and SNF percentages were also higher at the early LS compared to the late LS (*p* = 0.0001). Goat milk salts (minerals and trace elements) are mainly represented by K, Cl, Ca, and P ions. The principal influences of minerals in goat milk are the stage of lactation, breed, nutritional supply, and udder health [33,42]. Meanwhile, SNF is the sum of protein, lactose, and minerals [43]. Syd Jaafar et al. [44] did not find a relationship between the minerals in the diet and the minerals in goat’s milk (irrespective of goat genetics., i.e., Jamnapari, Boer, Saanen, or crossbreeds). They found a milk ash content of 0.67 to 0.87%. On the other hand, Idamokoro et al. [36] found significant differences in mineral composition by genotype and LS. Meanwhile, in the same study, the SNF values were only different between the LS of Nguni goats, but no differences were observed between breeds (Nguni, Boer, or nondescript goats) [36]. In our study, milk mineral contents (salts) were related to elements found with altitude (herbage minerals in the diet). In contrast to what was observed in Idamokoro et al.’s [36] study, Creole goat milk salts concentrations were higher at the early LS compared to that of the late LS. The differences in salts and SNF values observed by LS could be attributable to milk yield, as suggested by Iussing et al. [9].

The freezing point (FP) represents the composition of goat milk and the proportion of its components; its value has an inverse relationship with milk constituents [37]. Park et al. [33] reported goat milk FP in a range of −0.540 °C to −0.570 °C, whereas Strzalkowska et al. [37] reported a wider range of goat milk FP (−0.596 °C to −0.625 °C). Our observed FP values could mainly be related to variations in the concentration of salts and SNF by AR. A similar tendency is observed with the concentration of lactose and protein, that is, the higher the concentration of these milk components, the lower the observed FP value. Therefore, goat milk FP in our study seems to reflect variations due to the agroecological region. Moreover, significant FP differences are observed by LS, meaning that the early lactation FP was lower than the late lactation FP. The content of fat is an important factor that determines FP. At the lactation peak (beyond the 3rd month of lactation), lactose and protein have diminished compared to fat. Therefore, in this lactation period, FP is prone to increase [45]. In this research, we analyzed the goat milk in the first 70 days of lactation, and it is probably that differences in milk FP at early (10th day) and late (70th day) lactation could be due to an early weaning of the kid (at 21 days in farm conditions), where goatling lactation is not required. The SNF values support the observation that goat milk constituents had diminished by the late lactation period.

Goat milk pH is an important technological feature for cheese processing. Goat milk with a higher pH has a longer rennet clotting time [13]. Park et al. [33] reported a goat milk pH range of 6.5 to 6.8. In this study, we encountered constant mean values of 6.67–6.69, without differences caused by different altitude conditions. This pH value is in line with the goat milk pH ranges reported. Conversely, significant differences were observed between the pH at early (6.63) and late (6.66) LS. It has been reported that milk protein content is the leading factor affecting milk pH, especially casein content [46]. Caseins are peptides that form the micellar structure that binds and transports calcium phosphate. The a and b caseins form the inner structure of the micelle, while the k-casein forms the outer structure, with the glycosylation moiety. The glycosylation pattern of the casein micelle partly determines the mild acidic nature of milk under physiological conditions [47]. Considering that 80% of goat milk protein is casein [48], it is not surprising these peptides could determine milk pH. Nevertheless, in our study, protein differences between LS are minimal, although it reflects the pH obtained (less acidic pH at the late LS compared to the early LS). In agreement with this observation, Kuchtik et al. [39] observed that total milk protein had a positive correlation with titratable acidity.

Fat percentage in goat milk is variable, and it is determined by the breed (genetic background), lactation stage, and season, but mainly by feeding strategies [6,38]. The normal fat percentage reported is around 3.8%, with a range of 3.35% (Saanen breed) to 4.61% (Nubian breed) [6,33]. Moreover, Thakore and Jain [49] reported a goat milk fat of 5.2%. In our study, no differences were observed between lowland, highland, or mountain ARs (with an average range of 4.38% to 4.6%). In agreement with our results, Žan et al. [50] reported no differences between the fat content of goat milk in highland (630 masl, 3.77%) or mountain (1075 masl, 3.36%). In another study, Barlowska et al. [13] observed that milk fat was higher at mountain altitudes (3.91%, 500–1000 masl) compared to upland areas (3.29%, 300–500 masl), in contrast with our results. It is important to state that in the study of Barlowska et al. [13], the goat population was raised only for milk production and had concentrate feed supplementation, mainly in the winter season. The goat population that was studied in this work was raised mainly for kid production, the milk was a supplementary product. Further, no concentrate feed supplementation was added to the goats’ diets. The fat percentage by LS was higher at late lactation compared to that of early lactation (Table 1). Similar results were observed by Yilmaz et al. [49] in sheep milk, and Barlowska et al. [13] in goat milk. In the latter work, they showed that in the last stage of lactation (autumn) the milk yield decreases as the fat content increases (a negative correlation was observed). In this work we did not measure milk yield, however, milk yield probably decreases by the 70th day of lactation, thus increasing the fat percentage.

The principal FAs reported in goat milk are palmitic (C16:0), oleic (C18:1n9c), myristic (C14:0), stearic (C18:0), and lauric (C12:0) acids [33]. In this study, palmitic acid (C16:0; 28.5%), oleic acid (C18:1n9c; 17.8%), capric acid (C10:0; 10.9%), myristic acid (C14:0; 10.9%), and stearic acid (C18:0; 8.3%) were found in Creole goat milk as the principal milk FAs, without significant differences between ARs. Similar results were observed in Polish white improved goats [37], and in Saanen and Alpine breeds at highland (615–630 masl) and mountain (1060–1075 masl) altitudes [50]. It is of interest to state that under different altitude conditions (that determine the composition of herbs, forbs, shrubs, and other plants), we did not find evidence of differences in specific FA percentages, except for C14:1. The level of this unsaturated FA was higher in the milk from the mountain AR compared to that of the highland AR. The lowland C14:1 FA content was similar to that at the mountain and highland altitudes. Kondyli et al. [51] found C14:1 FA values of 0.58 to 0.67 g/100 g in the milk of the native goat breed of Greece, which is settled at a semi-mountainous altitude (600–800 masl). This value was higher than the C14:1 FA value in our work at a similar altitude (0.36 g/100 g, highland 853.3 masl). These differences could be related to the concentration of this FA in the vegetal material, which had a direct influence on the milk concentration of C14:1. In accordance with this statement, it was reported that grassland located in mountain areas (beyond 1000 masl) was a source of higher PUFA, MUFA, and CLA levels in sheep milk, compared to the grassland located at 500 masl [52].

The mid-chain FAs, mainly C6:0, C8:0, and C10:0, are of special interest due to their therapeutic activities (ease of human digestion and energy availability) [53], and it has been reported that milk of natural grazing goats has enriched mid-chain FA compared to that of goats with concentrate–forage diets [39]. Differences by LS were observed in C8:0 and C10:0 FA, where the late LS percentages were higher than those at the early LS. Capric FA was one of the most abundant FA encountered in goat milk, so differences observed in this FA are reflected in SCFA concentration, where late LS values were significantly higher than early LS values. Meanwhile, C18:2n6t FA was lower at the late LS compared to that of the early LS. Similar tendencies in the same fatty acids were observed in the milk of brown short-haired goats on an organic farm in the Czech Republic [39]. Contrary to our results, Strzalkowska et al. [37] observed the opposite tendency, i.e., C8:0 and C10:0 milk FA were significantly lower as lactation progressed, whereas C18:2 augmented as lactation progressed. On the other hand, Kondyli et al. [51] found no differences between C8:0, C10:0, or CLA content in indigenous goat milk in spring or summer. The observed differences between studies are associated with intrinsic characteristics, such as breed, but mainly with diet. In this regard, the plant’s phenological state could determine the content of FA and its proportion in milk [51]. Furthermore, this observation is compatible with the notion that linoelaidic fatty acid (C18:2n6t) was higher at the early LS. It was reported that low dietary crude protein and high dietary lignin could inhibit rumen FA biohydrogenation (BH). This partial BH inhibition promotes the accumulation of trans-6 to trans-11 octadecenoic acids (which are formed during BH of dietary alfa-linoleic, linoleic, and gamma linolenic acids) [54]. It is noteworthy that CLA values did not differ significantly between AR or LS (Table 2), suggesting that variations in C18:2n6t levels had very little effect on the CLA profile. Additionally, the meat-type profile of the Creole goats of northeast Mexico could contribute to the CLA value observed in their milk.

In this study, total milk SFA demonstrated average values of 76–78%, MUFA values ranged between 20–22%, and, finally, PUFA values ranged between 1.7–1.36%. No differences were observed by AR or LS. Slight differences were reported by Zervas and Tsiplakou [38]. They reported an average of 74% for SFA and 21% for MUFA. On the other hand, their PUFA-reported values are quite different from our results. They reported PUFA goat milk values of 5%. In agreement with the latter, Žan et al. [50] found PUFA milk values of 3.73% and 3.24% at highland and mountain altitudes, respectively, and Strzalkowska et al. [37] found an increase in PUFA goat milk concentration as lactation progressed (in the range of 2.82–4.73%). The PUFA content in goat milk is positively related to vegetation diversity, especially to Fabaceae and woody species [54]. The vegetation eaten by the grazing Creole goats in semiarid rangeland probably has a lower PUFA content compared to other studies. Additionally, it is important to state that the Creole goats in this study did not receive supplementation feeding, and the genotypic (breed) characteristics cannot be ruled out since Creole goats in northeast Mexico are raised mainly for meat purposes and not for milk purposes. Taken together, the above observations let us hypothesize that both feed conditions and Creole breed characteristics (of a meat type) are aspects that define the milk FA profile.

It is stated that lipidic indices are important values that report human cardiovascular health risks [26]. The mean HH values ranged between 0.48–0.57 without differences in AR nor LS. In agreement with us, Pietrzak-Fiecko and Kamelska-Sadowska [55] reported similar results when comparing the nutritional value of human milk with other mammals’ milk. They reported the following HH hierarchy: human > mare > cow > goat > sheep (1.67, 1.65, 0.83, 0.59, and 0.44, respectively). On the other hand, Osmari et al. [25] showed an HH average of 0.75 in the milk of goats fed with sorghum or maize silage. These authors indicated that high values for HH are desirable considering the effects of fatty acid on cholesterol metabolism. Bodnár et al. [56] described the HH as the relationship between hypocholesterolemic fatty acids (cis-C18:1 and PUFA) and hypercholesterolemic fatty acids (lauric, myristic, and palmitic acids). Therefore, a high HH index contains a greater number of hypocholesterolemic fatty acids that are considered health-promoting for humans [55]. Our results demonstrated that Creole goat milk fulfills the HH index reported regardless of AR.

Conversely to what is observed for HH, low values of AI and TI could reduce the risk of human coronary heart disease [25]. In this study, AI showed mean values range of 3.18–4.24 without differences for AR nor LS. In agreement with us, Pietrzak-Fiecko and Kamelska-Sadowska [55] reported a goat milk AI index similar to our results. They also reported the following mammal AI milk hierarchy: sheep > goat > cow > human > mare (4.21, 3.17, 2.37, 1.12, and 1.11, respectively). Cui et al. [57] reported that AI was significantly lower at the middle and higher altitudes compared to that of lower altitudes when evaluating yak milk. This same tendency was observed in our study, without significant differences. Opposite to that which was observed for AI, the highland AR milk had a significantly lower TI value (1.90) compared to that of the lowland AR (2.34). The differences regarded could be explained by the lower thrombogenic fatty acids (C18:0 and C18:1) contents in the goat milk of the highland AR. In agreement with this observation, Cui et al. [57] reported that saturated fatty acids decreased with altitude. The observed TI range (2.0–2.34) was slightly above what was reported by Pietrzak-Fiecko and Kamelska-Sadowska [55]. They reported a goat milk TI of 2.06. On the other hand, Osmari et al. [25] reported a higher goat milk TI (2.85 to 3.10) when goats are fed with mulberry hay or maize silage, respectively. These observations highlight the importance of feed on the milk FA profile. In this study, the Creole goat milk without supplementation had a TI that could be less thrombogenic than goats with silage-supplemented feeding.

## 5. Conclusions

The physicochemical characteristics of Creole goat milk are dependent on the agroecological region of northeast Mexico, with semiarid rangeland as the only feeding source of nutrients. The chemical composition of this goat milk makes it suitable for human consumption, given that nutritional parameters agree with what has been previously reported. One of the best parameters studied in goat milk is the fatty acids profile. In this study, we demonstrated that Creole goat milk has a similar HH index, but slightly higher AI and TI. Goat feed supplementation could improve milk quality (e.g., a higher milk CLA content) as it can be viewed as a profitable dairy product. The meat-type phenotypical characteristics of the Creole goat could also influence milk composition. The goat pastoralist conditions of northeast Mexico tend toward poverty, so the implementation of new strategies that alleviate this social condition could be a solution.

## Figures and Tables

**Table 1 animals-13-01738-t001:** Coordinates and altitudes of the goat farm locations in the three agroecological regions of northeast Mexico.

AR	Location	Coordinates		Altitude masl
		N	W	
Lo	Comas altas	25°02′33″	98°26′14″	71
	Guadalupe	25°07′11.9″	98°46′58.2″	101
	Candido Aguilar	25°06′44.4″	98°39′46.3″	89
Hi	Jose Ma. Morelos	23°30′20.3″	99°22′16.1″	767
	Jose Ma. Morelos	23°20′22.8	99°22′24.8″	772
	San Juanito	23°27′19.2″	99°24′40.6″	800
Mo	Alvaro Obregon	23°10′42.2″	99°41′06.6″	1406
	Alvaro Obregon	23°10′26.0″	99°41′40.4″	1379
	Rancho nuevo	23°02′34″	99°32′43″	1143

AR: agroecological region; N: north; W: west; Lo: lowland; Hi: highland; and Mo: mountain.

**Table 2 animals-13-01738-t002:** Creole goat milk composition from three agroecological regions and two stages of lactation in northeast Mexico.

Chemical Composition	AR			SEM	LS		SEM	*p*-Value		
	Lo	Hi	Mo		Ea	La		AR	LS	Interaction AR × LS
Density kg/m^3^	1033.8 ^b^	1032.5 ^b^	1036.8 ^a^	1.207	1035.2	1034.2	1.208	0.016	0.449	0.289
Fat, %	4.38	4.53	4.6	0.184	4.07 ^b^	5.07 ^a^	0.167	0.687	0.0001	0.0001
Lactose, %	5.19	5.1	5.25	0.05	5.21	5.2	0.051	0.081	0.518	0.205
Protein, %	3.84 ^a^	3.43^b^	3.58 ^b^	0.102	3.74	3.69	0.102	0.014	0.060	0.068
Freezing point, °C	−0.652 ^b^	−0.622 ^a^	−0.654 ^b^	0.008	−0.679 ^b^	−0.631 ^a^	0.007	0.006	0.0001	0.398
pH	6.67	6.68	6.69	0.012	6.63 ^b^	6.66 ^a^	0.011	0.399	0.0001	0.0001
Salts, %	0.76 ^a^	0.73 ^b^	0.76 ^a^	0.009	0.79 ^a^	0.73 ^b^	0.736	0.011	0.0001	0.726
SNF, %	9.96 ^a^	9.47 ^b^	9.89 ^a^	0.111	10.31 ^a^	9.55 ^b^	0.091	0.003	0.0001	0.702

AR: agroecological region; LS: lactation stage; Lo: lowland; Hi: highland; Mo: mountain; Ea: early lactation (10th day); La: late lactation (70th day); SNF: solid not fat; and SEM: standard error mean. Different superscript letters mean statistical difference (*p* < 0.05).

**Table 3 animals-13-01738-t003:** Mean fatty acids concentrations (% of total FA, g/100 g) and lipidic indexes for Creole goat milk from three agroecological regions and two stages of lactation in northeast Mexico.

Fatty Acid	AR			SEM	LS		SEM	*p*-Value		
	Lo	Hi	Mo		Ea	La		AR	LS	AR × LS
C4:0	1.74	1.92	1.92	0.293	1.62	1.79	0.233	0.866	0.142	0.699
C6:0	3.81	3.82	3.27	0.475	3.19	3.51	0.364	0.616	0.054	0.303
C8:0	4.08	4.06	3.53	0.378	3.44 ^b^	3.81 ^a^	0.329	0.532	0.049	0.307
C10:0	11.07	11.0	10.61	1.002	9.52 ^b^	11.39 ^a^	0.783	0.943	0.033	0.079
C11:0	0.19	0.11	0.14	0.048	0.20	0.12	0.049	0.491	0.462	0.717
C12:0	5.3	4.69	4.38	0.349	4.54	4.83	0.318	0.205	0.523	0.749
C13:0	0.17	0.11	0.13	0.025	0.13	0.14	0.022	0.284	0.9	0.576
C14:0	9.88	11.92	10.94	0.597	11.19	11.23	0.62	0.103	0.518	0.262
C14:1	0.40 ^ba^	0.36 ^b^	0.65 ^a^	0.073	0.47	0.48	0.065	0.038	0.993	0.222
C15:0	1.70	1.35	1.62	0.152	1.46	1.66	0.119	0.300	0.425	0.116
C15:1	0.47	0.41	0.54	0.043	0.49	0.50	0.043	0.177	0.512	0.451
C16:0	26.57	29.71	27.88	0.948	28.39	27.84	0.811	0.117	0.861	0.525
C16:1	0.88	0.61	0.87	0.080	0.75	0.78	0.067	0.073	0.528	0.308
C17:0	1.27	1.13	1.31	0.096	1.31	1.26	0.083	0.442	0.192	0.120
C17:1	0.47	0.33	0.41	0.043	0.39	0.47	0.077	0.148	0.082	0.447
C18:0	8.47	7.73	8.89	1.059	8.82	8.03	0.834	0.762	0.734	0.413
C18:1n9t	1.68	1.34	1.24	0.133	1.52	1.50	0.213	0.089	0.086	0.064
C18:1n9c	17.92	17.13	18.48	1.019	18.48	17.63	1.006	0.764	0.629	0.581
C18:2n6c	0.14	0.18	0.27	0.056	0.23	0.17	0.070	0.285	0.771	0.719
C18:2n6t	0.94	0.94	1.14	0.141	1.16 ^a^	1.07 ^b^	0.116	0.546	0.016	0.252
C18:3n6	0.27	0.13	0.28	0.125	0.15	0.31	0.100	0.686	0.439	0.327
Others	2.47	1.00	1.55	1.125	2.46	1.37	1.164	0.676	0.710	0.287
SFA	76.78	78.61	76.23	1.155	76.31	77.03	1.127	0.382	0.433	0.503
MUFA	21.85	20.12	22.06	1.057	22.12	21.39	1.037	0.431	0.524	0.522
PUFA	1.36	1.26	1.70	0.192	1.55	1.57	0.175	0.285	0.169	0.722
SCFA	20.71	20.81	19.35	1.845	17.78 ^b^	20.51 ^a^	1.467	0.830	0.016	0.161
MCFA	47.35	50.79	48.91	1.252	49.36	49.36	1.212	0.215	0.768	0.578
LCFA	31.92	28.38	31.72	2.200	32.35	30.12	1.955	0.503	0.304	0.168
CLA	1.08	1.12	1.41	0.128	1.40	1.25	0.133	0.162	0.800	0.449
HH	0.48	0.57	0.54	0.033	0.54	0.53	0.034	0.210	0.768	0.446
AI	4.24	3.18	3.36	0.287	3.34	3.64	0.308	0.060	0.572	0.367
TI	2.34 ^a^	1.90^b^	2.00 ^ab^	0.095	1.99	2.09	0.108	0.021	0.547	0.253

AR: agroecological region; LS: lactation stage; Lo: lowland; Hi: highland; Mo: mountain; Ea: early lactation (10th day); and La: late lactation (70th day). Values within rows with different superscript letters are significantly different from each other (*p* < 0.05). Abbreviations: SFA, total saturated fatty acids; MUFA, total monounsaturated fatty acids; PUFA, total polyunsaturated fatty acids; SCFA: total short-chain fatty acids (∑ C4, C8, C6, C10); MCFA: total medium-chain fatty acids (∑ C11, C12, C13, C14, C14:1, C15, C15:1, C16, C16:1, C17, C17:1); LCFA, total long-chain fatty acids (∑ C18, C18:1n9t, C18:1n9c, C18:2n6c, C18:2n6t, C18:3n6, others); CLA: conjugated linoleic acids (∑ C18:2n6c, C18:2n6t); HH, hypo:hypercholesterolemic acids; AI, atherogenic index; and TI, thrombogenic index.

## Data Availability

The data that support this study are presented as Appendix A.

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
