# Peer review of "Milk Composition of Creole Goats Raised at Different Altitudes in an Extensive Production System in Northeast Mexico"

_animals, 2023, doi:10.3390/ani13111738_

Round 1
Reviewer 1 Report
It is an exciting study with several practical implications. However, it would be interesting to know some lipid quality indices such as the atherogenicity index, index thrombogenicity, and hypocholesterolemic/hypercholesterolemic index of creole goat's milk of crucial importance in human health. It is also necessary to clarify some points in the methodology and improve the discussion by emphasizing how your results can affect human health and the manufacture of goat's milk byproducts.

Reviewer 3 Report
Comments and Suggestions for Authors
The manuscript is clearly written and structured, with adequate method descriptions.
However, the presentation of the results needs to be revised for more clarity and the discussion has to be summarized and focused on the interest of the results obtained, avoiding redundant sentences. References should be reduced to the more significant ones.
All comments are listed below point by point and some detailed comments are reported in the attached PDF file.
Abstract
The experimental theses are not clearly explained in the abstract. Please report in detail the three agroecological regions and their respective altitudes.
Introduction
The objective of the work should be explained at the end of the introduction.
The impact of this study on Creole goat farming in Mexico should be highlighted.
Materials and Methods
L 70-74 Please, check the altitude and annual temperature of the three agroecological regions.
In all the text, it is important to define the three theses as agroecological regions because they are different not only in the altitude but in the environmental characteristics as a whole.
Results
Review Table 1 (see attached PDF file).
Discussion
The discussion has to be summarized and focused on the interest of the results obtained, avoiding redundant sentences. Are the observed differences in milk characteristics important technological characteristics for cheese processing?
Some detailed comments are reported in the attached PDF file.
Conclusions
In my opinion, the conclusion should report the impact of these findings on Creole goat farming in Mexico.
The interest in the composition of Creole goat milk should be highlighted. The obtained results could suggest how to improve the quality of Creole Goat milk.

Please, review English spelling especially the present or past tense of verbs.
Reviewer 4 Report
In general, all parts of the article are written in understandable language and the transitions between sections are well organized. I would like to congratulate the authors for the good quality of the article, the literature reported used to write the paper, and for the clear and appropriate structure. The manuscript is well written, presented and discussed, and understandable to a specialist readership.
In general, the organization and the structure of the article are satisfactory and in agreement with the journal instructions for authors. The subject is adequate with the overall journal scope.
The work shows a conscientious study in which a very exhaustive discussion of the literature available has been carried out. The introduction provides sufficient background, and the other sections include results clearly presented and analyzed exhaustively.
In order to further improve the quality of the paper, an overall check of the English language is recommended.
So, I recommend the acceptance of the paper after a minor revision.
In order to further improve the quality of the paper, an overall check of the English language is recommended.
So, I recommend the acceptance of the paper after a minor revision.
Author Response
Tamaulipas, Mexico. May 7, 2023
Dear Reviewer
On behalf of the authors of the manuscript “Creole Goat Milk Composition Raised at Different Altitudes in an Extensive Production System in Northeast Mexico”, I state that the grammar language of the manuscript was revised.
I hope that the corrections that we made accomplish all your requirements. We appreciate the revisions and suggestions for the benefit of the manuscript.
Sincerely
Dr. Jorge Alva-Pérez
Corresponding author.
Round 2
Reviewer 3 Report
The work is improved after revision and some aspects have been cleared.
Some detailed comments are reported in the attached PDF file.
Review Table 2 (see attached PDF file).
L 24-25 and L 79-80 Please, check the altitude of the three agroecological regions. The definition of the three agroecological regions needs to be clarified.
It is not clear how the agro-ecological regions have been defined based on the altitude: lowland (50-300 masl); highland (400-2800 masl) and mountain (300-3200 masl). The reported ranges of altitude are very large and overlap between highlands and mountains.
For this reason, it seems incorrect to define milk according to altitude but rather according to agro-ecological regions. It would be important to know the altitude of the nine farms selected from the three different agro-ecological sites (Line 98) and the altitude range where the goats actually grazed.
If this point is not clarified, it seems that there are few differences between the three agro-ecological regions and the few differences found in milk, in particular in FA, between the three agro-ecological areas could be due to the fact that the three areas are not very different in terms of altitude range, average temperature, precipitation range (Line 82-84) and vegetation (Line 89).
In all the text, it is important to define the three theses as agroecological regions because they are different not only in the altitude but in the environmental characteristics as a whole.

Check the spelling of the English language.
